# Risk Factors Predisposing to Angina in Patients with Non-Obstructive Coronary Arteries: A Retrospective Analysis

**DOI:** 10.3390/jpm12071049

**Published:** 2022-06-27

**Authors:** Oskar Wojciech Wiśniewski, Franciszek Dydowicz, Szymon Salamaga, Przemysław Skulik, Jacek Migaj, Marta Kałużna-Oleksy

**Affiliations:** 1Faculty of Medicine, Poznan University of Medical Sciences, 10 Fredry Street, 61-701 Poznan, Poland; franekdyd@gmail.com (F.D.); szymon.salamaga@gmail.com (S.S.); przemekskulik@wp.pl (P.S.); 21st Department of Cardiology, Poznan University of Medical Sciences, 1/2 Dluga Street, 61-848 Poznan, Poland; jacek.migaj@skpp.edu.pl (J.M.); marta.kaluzna@wp.pl (M.K.-O.)

**Keywords:** angina pectoris, cardiac syndrome X, coronary angiography, coronary artery disease, inflammation, lipids, lipid profile, microvascular dysfunction, non-obstructive coronary artery disease

## Abstract

No hemodynamically significant atherosclerotic plaques are observed in up to 30% of patients reporting angina and undergoing coronary angiography. To investigate risk factors associated with non-obstructive coronary artery disease (NOCAD), we analyzed the medical records of, consecutively, 136 NOCAD subjects and 128 patients with significant stenosis in at least one coronary artery (the OCAD group). The blood concentrations of the TC (4.40 [3.78–5.63] mmol/L vs. 4.12 [3.42–5.01] mmol/L; *p* = 0.026), LDL-C (2.32 [1.80–3.50] mmol/L vs. 2.10 [1.50–2.70] mmol/L; *p* = 0.003), non-HDL-C (2.89 [2.29–4.19] mmol/L vs. 2.66 [2.06–3.39] mmol/L; *p* = 0.045), as well as the LDL-C/HDL-C ratio (1.75 [1.22–2.60] vs. 1.50 [1.10–1.95]; *p* = 0.018) were significantly increased in the NOCAD patients compared to the OCAD group due to the lower prevalence and intensity of the statin therapy in the NOCAD individuals (*p* < 0.001). Moreover, the abovementioned lipid parameters appeared to be valuable predictors of NOCAD, with the LDL-C (OR = 1.44; 95%CI = 1.14–1.82) and LDL-C/HDL-C (OR = 1.51; 95%CI = 1.13–2.02) showing the highest odds ratios. Furthermore, multivariable logistic regression models determined female sex as the independent risk factor for NOCAD (OR = 2.37; 95%CI = 1.33–4.20). Simultaneously, arterial hypertension substantially lowered the probability of NOCAD (OR = 0.21; 95%CI = 0.10–0.43). To conclude, female sex, the absence of arterial hypertension, as well as increased TC, LDL-C, non-HDL, and LDL-C/HDL-C ratio are risk factors for NOCAD in patients reporting angina, potentially as a result of poor hypercholesterolemia management.

## 1. Introduction

Coronary artery disease (CAD), one of the most common cardiovascular (CV) diseases, is the leading cause of death worldwide [1]. Interestingly, the results of large multi-center studies showed the absence of hemodynamically significant lesions in at least one coronary artery in 20–30% of symptomatic patients, despite the presence of typical chest pain [2,3,4]. According to some authors, traditional CAD risk factors, such as arterial hypertension, diabetes mellitus (DM), dyslipidemia, or a family history of early-onset CV disease, account for less than 20% of angina in patients without significant lesions [5]. These findings led to the introduction of non-obstructive CAD (NOCAD), historically referred to as cardiac syndrome X (CSX). CSX is defined by angina-like chest discomfort and the presence of electrocardiographic ST/T-segment abnormalities despite angiographic visualization of normal coronary arteries [6].

NOCAD is more prevalent in women, especially those of peri- or postmenopausal age [7,8]. A study of 32,856 patients with CAD suspicion revealed that 23.3% of women and only 7.1% of men had non-obstructive coronary arteries as determined by the standard angiography [9]. In addition, there are unique risk factors for NOCAD in women, such as pregnancy-related disorders, autoimmune dysfunction, chronic inflammation, and psychological risk factors [10].

The diagnosis of NOCAD in patients with chest pain and non-obstructive coronary arteries is made by exclusion, bearing in mind non-cardiac causes of musculoskeletal, gastrointestinal, psychiatric, or pulmonary origin [7,11]. The etiology of NOCAD is unclear and multifactorial. Nevertheless, the pathogenesis can be attributed to two highly supported hypotheses of coronary microvascular dysfunction (CMD) and abnormal cardiac pain sensitivity; these may contribute to NOCAD either collectively or separately [12,13].

The first theory of CMD assumes that the chest pain results from myocardial ischemia due to a dysfunction of small blood vessels (<500 μm in diameter), which are not visible via standard coronary angiography [14]. Patients with NOCAD had a high prevalence of CMD irrespective of sex. Sara et al. established that 64% of the NOCAD patients were evaluated to have CMD [15], while a recent meta-analysis by Mileva et al. revealed a 41% prevalence of CMD among NOCAD patients [16]. The clinical manifestation of CMD may be associated with inappropriate, extensive vasoconstriction and/or the impaired relaxation of arterioles and prearterioles; this reduces the supply of blood to cardiomyocytes [4]. Physiologically, human coronary circulation is regulated by nitric oxide (NO) and a reduced concentration of hydrogen peroxide (H_2_O_2_) as an endothelium-derived hyperpolarizing factor [13]. Multiple cardiovascular risk factors (e.g., aging, hypertension, obesity, and smoking) may provoke an increase in H_2_O_2_ and a reduction in NO production. As a result of these biochemical changes, vasoconstriction as well as pro-inflammatory and pro-thrombotic states are favored [13]. NOCAD patients were reported to have undermined vasodilation in response to endothelium-dependent and endothelium-independent hyperemic stimuli, higher ischemic metabolite production after tachycardia, greater thallium scan defects, as well as more heterogeneous blood flow [13,17,18,19,20].

Conversely, a decreased tolerance for cardiac pain may be the pathophysiological background for NOCAD. In addition, these patients are much more likely to suffer from anxiety and panic disorders, which could be responsible for angina-like symptoms [21,22]. Moreover, the hypersensitivity to right atrial pressure and volume alterations may play a role as well [23].

Nevertheless, the pathogenesis of NOCAD requires further research, which may help develop more specialized treatment options. Currently, the management of NOCAD is based on conventional anti-anginal therapy, which includes anti-ischemic and analgesic medications, non-pharmacological procedures, and lifestyle modification. Unfortunately, it is not always effective [24].

In the past, the diagnosis of NOCAD was frequently underestimated and considered benign [25]. However, recent studies demonstrated that all-cause mortality and the rate of CV events, such as myocardial infarction (MI), stroke, hospitalization for heart failure, or cardiac death, are higher among patients with NOCAD than in that of the general population [26,27]. Furthermore, NOCAD contributes to a significant decrease in the quality of life and restricts daily activities. Moreover, the diagnostic process is often long and expensive (estimated at 1 million USD) because many patients undergo coronary angiography several times [28].

A better understanding of NOCAD, including the CV profiles of the NOCAD patients, may improve the eligibility criteria for coronary angiography. In addition, reducing the number of coronary angiographies in patients with non-obstructive coronary arteries might protect them from periprocedural complications and decrease overall treatment costs.

The importance of a non-obstructive plaque and the emphasis that this stage of coronary atherosclerosis can no longer be ignored is the most reasonable today, especially in treating CV risk factors and wide-ranging prevention [29].

The study aimed to determine factors associated with angina in patients with NOCAD. A detailed description of such factors could prove helpful in the diagnosis, treatment, and prognosis assessment of patients with NOCAD.

## 2. Materials and Methods

We analyzed the medical records of 264 patients reporting angina (median age 68.00 [60.00–73.50] years; 37% females) who underwent coronary angiography in the 1st Department of Cardiology of Poznan University of Medical Sciences between January and April 2019. Participants with acute coronary syndromes, acute infectious diseases, myocardial bridges, Prinzmetal’s angina, and a history of heart transplantation were excluded.

All angiograms were performed and evaluated by an experienced interventional cardiologist who stated the percentage of stenosis and qualified participants to the study or the control group. The study group (NOCAD) consisted of 136 patients without hemodynamically significant lesions in any coronary artery, while the control group (OCAD) comprised 128 individuals with hemodynamically significant lesions in at least one coronary artery. Detailed characteristics of the population are summarized in Table 1.

All data gathered in medical records were analyzed, including anamnesis, clinical profile, past medical history, family history, pre-hospital treatment, and details of laboratory and echocardiographic parameters. Cardiovascular diseases included within family history were MI, CAD, stroke, transient ischemic attack (TIA), arterial hypertension, and premature sudden cardiac death (defined as unexpected death due to a cardiac cause within 24 h of being seen well in young adults under 45 years old). Alcohol consumption was classified into one of three categories: none, occasionally (less than 8 drinks per week for women or less than 15 drinks per week for men), and frequently (8 or more drinks per week for women or 15 or more drinks per week for men). The transthoracic two-dimensional M-mode echocardiography was performed by one specialist using the same device in all patients, and the following parameters were assessed: ejection fraction, left atrium diameter, left ventricle end-diastolic diameter, right ventricle diameter, posterior wall thickness, and interventricular septum thickness. Laboratory tests performed at admission (complete blood count (CBC), lipid profile, fasting plasma glucose level, serum creatinine level, high-sensitive C-reactive protein, and cardiac troponin I) as well as the results of electro- or echocardiography were available in all patients. Additionally, inflammatory and lipid ratios, such as neutrophil/lymphocyte (NLR), lymphocyte/monocyte (LMR), as well as LDL-C/HDL-C and non-HDL-C/HDL-C, were calculated. BMI was calculated by dividing body mass in kilograms by the square of height in meters.

Because atorvastatin was the most common statin in both groups, doses of other statins were converted into approximate equivalent atorvastatin doses according to the formula suggested by Livingston et al. [30]. To do so, rosuvastatin doses were multiplied by 2.0, while simvastatin doses were divided by 2.0.

The Shapiro–Wilk test was applied to evaluate the normal distribution of variables. Categorical variables were shown as numbers and percentages (presented in parentheses). Then, they were compared using the Chi-squared test. Continuous variables were displayed as median (q1–q3) and compared using the Mann–Whitney U test, given the lack of normal distribution. Finally, correlation analyses were performed using the Spearman correlation.

Additionally, univariable and multivariable logistic regression models were created and analyzed. In both, the dependent variable was the lack of hemodynamically significant narrowing of any coronary artery. Model variables with a likelihood ratio *p*-value (pLR) < 0.05 in the abovementioned univariable logistic regression analysis were initially chosen for multivariable logistic regression. Then, parameters showing high correlations with a lot of variables (age, TC) were excluded. All remaining parameters were assessed in the analysis of representatives and assigned to specific classes. Because LDL-C, LDL-C/HDL-C, non-HDL-C, non-HDL-C/HDL-C, and the approximate atorvastatin dose all appeared colinear, five different multivariable logistic regression models were created to compare the influence of each lipid-related parameter. In the interaction analysis, no significant interactions were observed for such prepared models. V-cross validation was performed when building the models. The goodness of fit was confirmed with Wald’s test *p*-value < 0.001 and a high area under the curve (AUC) of about 0.76.

Statistical analysis was conducted using Statsoft Statistica 13.3 software (TIBCO Software Inc., Palo Alto, CA, USA, 2017). *p*-value < 0.05 was considered statistically significant.

Additional analyses within the OCAD group regarding patients with obstruction in one or multiple coronary arteries were also performed. The results are presented in the Appendix A.

## 3. Results

### 3.1. Study Population Characteristics

Detailed study population characteristics are shown in Table 1. Interestingly, the percentage of females was significantly higher in the NOCAD group (45%) than in the OCAD group (29%, *p* = 0.007). In both groups, most patients lived in the countryside and were either overweight or obese. Merely one-fourth of the patients in both groups had a normal body mass index (BMI). There was no statistically significant difference in the age of the patients between groups.

Arterial hypertension and DM were much more prevalent in the OCAD group than in the NOCAD group (91 vs. 63%, *p* < 0.001, and 36 vs. 24%, *p* = 0.027, respectively). However, no statistically significant differences were found between groups regarding heart failure or positive family history of CV diseases. Of note, 10% of the NOCAD group and 42% of the OCAD group had prior MI (*p* < 0.001), while 9 and 62% (*p* < 0.001) of patients underwent prior percutaneous coronary intervention (PCI) in the NOCAD and OCAD groups, respectively.

Most patients denied smoking or alcohol abuse. We revealed no significant associations between the use of stimulants and NOCAD or OCAD.

### 3.2. Laboratory Parameters

Concentrations of total cholesterol (TC), low-density lipoprotein cholesterol (LDL-C), and non-high-density lipoprotein cholesterol (non-HDL-C) were significantly increased in the NOCAD group as compared to the OCAD group (Table 2). Furthermore, an increase in the LDL-C/HDL-C ratio was observed in the NOCAD group. However, no significant differences in the HDL-C and triglycerides (TG) levels between the groups were identified.

Moreover, no significant alterations of the white blood count (WBC), different leukocyte subtypes, and platelets were present. Nevertheless, there was a trend for increased WBC, neutrophil count, and neutrophil/lymphocyte ratio (NLR) values in the NOCAD group compared to the OCAD group (Table 2). Moreover, the high-sensitive C-reactive protein (hs-CRP) concentration was slightly higher in the OCAD group, although it did not reach statistical significance.

Interestingly, the cardiac troponin I (cTnI) analysis showed that patients with non-obstructive coronary arteries presented with significantly lower cTnI values than patients from the OCAD group, indicating that intensified chronic troponin leakage may be associated with stenosed coronary arteries (Table 2).

### 3.3. Echocardiographic Parameters

An assessment of the transthoracic echocardiography parameters demonstrated a lower left ventricular ejection fraction (LVEF) in the OCAD group (Table 2). Other parameters did not differ significantly between the groups.

### 3.4. Univariable Regression Model

The univariable logistic regression analysis results (Figure 1, Appendix A) revealed that female sex is the key factor associated with NOCAD in patients reporting angina, doubling the likelihood of NOCAD (OR = 2.00; 95%CI = 1.20–3.33). Furthermore, most lipid parameters also elevated the chance of NOCAD, with LDL-C (OR = 1.44; 95%CI = 1.14–1.82) and LDL-C/HDL-C (OR = 1.51; 95%CI = 1.13–2.02) showing the highest odds ratios. On the other hand, arterial hypertension and DM decreased the odds ratio for NOCAD by 82% (OR = 0.18; 95%CI = 0.09–0.35) and 45% (OR = 0.55; 95%CI = 0.32–0.94), respectively.

### 3.5. Multivariable Regression Model

Each of the five distinct multivariable logistic regression models (Table 3) identified female sex as an independent factor associated with the incidence of NOCAD. Conversely, arterial hypertension independently decreased the likelihood of NOCAD. Moreover, LDL-C/HDL-C, non-HDL-C, and non-HDL-C/HDL-C were shown as independent NOCAD predictors, when used separately. Among them, LDL-C and LDL-C/HDL-C were described with the highest odds ratios. Notably, DM did not reach statistical significance to become an independent factor affecting the NOCAD occurrence.

### 3.6. Differences in Pharmacotherapy between OCAD and NOCAD Patients

Acetylsalicylic acid, angiotensin-converting enzyme inhibitors (ACE-I), angiotensin II receptor blockers (ARB), calcium channel blockers (Ca-blockers), and nitrates were used more frequently in the OCAD group when compared with the NOCAD group. Conversely, novel oral anticoagulants (NOAC) and vitamin K antagonists (VKA) were administered more frequently to the NOCAD patients compared to the OCAD individuals. No statistically significant difference in the use of beta-blockers, diuretics, metformin, mineralocorticoid receptor antagonists (MRA), and trimetazidine was observed between the two populations. The data are summarized in Table 4.

When it comes to lipid-lowering treatment (Table 5), statins were used statistically more often in the OCAD group, although most patients in both groups (OCAD vs. NOCAD: 83.20 vs. 54.69%; *p* < 0.001, respectively) took statins (atorvastatin, rosuvastatin, or simvastatin). Furthermore, after converting all statin doses into approximate equivalent atorvastatin doses, it turned out that patients with OCAD received more frequently higher statin doses (>20 mg/d), while the statin at a lower dose (≤20 mg/d) was more common in patients with NOCAD. The median approximate equivalent atorvastatin dose for the OCAD patients was 40 (20–40) mg/d and 20 (20–40) mg/d for the NOCAD group (*p* < 0.001). A fenofibrate addition to the lipid-lowering treatment was statistically more common in the OCAD group. However, there was no statistically significant difference in the treatment with ezetimibe between the groups. Additionally, it should be mentioned that none of the patients undergoing lipid-lowering therapy received proprotein convertase subtilisin/kexin 9 (PCSK-9) inhibitors.

## 4. Discussion

In our study, patients with NOCAD presented higher levels of TC and LDL-C (*p* < 0.05), and hence an elevated LDL-C/HDL-C ratio (*p* < 0.05), when compared to patients with hemodynamically significant coronary lesions. Significant differences in HDL-C or TG between both groups were not observed. The results of our investigation stand in contrast to most studies. There are many papers in which patients with NOCAD demonstrated a more favorable lipid profile than patients with OCAD [31,32]. As known, lower LDL-C and higher HDL-C levels are considered more favorable. Tselepis et al. showed that patients with microvascular angina presented raised HDL-C and a decreased LDL/HDL ratio than patients with OCAD [33]. Similar observations were made by Choo et al. who also observed increased HDL-C and declined LDL-C levels in patients with NOCAD [34]. Moreover, they found significantly lower levels of TG in the NOCAD group. Meanwhile, Ortega et al. showed that patients with NOCAD significantly less often presented dyslipidemia than patients with OCAD. Additionally, they confirmed that NOCAD patients had increased HDL-C and decreased TG levels [35]. The CRUSADE [36] and the APPROACH trials [37] are another two studies which suggest that patients without significant lesions were less likely to have dyslipidemia. NOCAD patients were found to possess fewer traditional risk factors than OCAD patients in these trials [36,37]. Finally, in a large meta-analysis including more than 120,000 patients, Pizzi et al. confirmed a pattern of decreased dyslipidemia rates among NOCAD patients [38].

Lipid parameters are of great importance in the pathophysiology of microvascular angina, which results from endothelial injury of small coronary arteries (<500 μm in diameter) [39]. Hypercholesterolemia is thereby one of the causes of endothelial dysfunction because it impairs the endothelium-mediated vasodilatory response [40,41]. Especially, LDL-C and lipoprotein a (Lp(a)) are well-known factors associated with a destructive impact on the endothelium [42]. In addition, many studies reported higher HDL-C levels in NOCAD patients, consistent with the cardioprotective effects of HDL-C. HDL-C levels negatively correlate with the intensity of systemic inflammation, a crucial component in atherosclerotic plaque formation and endothelial dysfunction [43].

Importantly, some lipid ratios, such as LDL-C/HDL-C, monocyte/HDL-C, or TG/HDL-C, are predictors of long-term mortality resulting from microvascular angina and CAD [44,45,46,47]. Therefore, a follow-up examination would be an objective for our further studies.

The results of the current literature directly oppose the results of our study. In a search for the justification of such unexpected results, we identified studies showing that patients without hemodynamically significant lesions in coronary angiography were treated less intensively and less often received statins than patients with significant stenoses [34,35,38]. The same observation also concerned other medications, including ACEI. Only 54.69% of the NOCAD patients vs. 83.20% of the OCAD patients received statins in our study. It appears that therapeutical decisions are more willingly made on behalf of angiography examination than on clinical presentation. As a result, patients with NOCAD remain undertreated and are thereby exposed to worse clinical outcomes. This hypothesis may explain a more deteriorated lipid profile observed in our study. The belief that microvascular angina is a benign condition seems entirely inaccurate. Indeed, it characterizes lower all-cause mortality and MACE rates (2.38 and 9.24% per year) than observed in OCAD patients (10.1 and 16.8% per year); however, NOCAD patients are still at high risk of death and MACE [31,38]. Not only do statins reduce cholesterol levels, but they also improve endothelium-dependent vasomotion, making them even more beneficial [48]. In addition, statins may decrease all-cause mortality in patients with myocardial infarction with non-obstructive coronary arteries (MINOCA) [34]. Each of these principles illustrates how optimal management of dyslipidemia is essential in NOCAD patients [49]. We noticed that a more impaired lipid profile in the NOCAD group was associated with inadequate statin therapy. Therefore, the higher intensity of lipid-lowering therapy in patients with OCAD may explain why this group showed statistically lower lipid levels than patients with NOCAD.

The neutrophil-to-lymphocyte ratio (NLR) is recognized as a marker of inflammation in cardiovascular diseases such as acute coronary syndrome [50,51], arterial hypertension [52], or stroke [53]. Patients diagnosed with CSX showed significantly higher NLR values than those reporting angina-like chest pain without coronary lesions [54]. Moreover, a higher NLR value was observed in CSX patients with impaired myocardial perfusion than patients with CSX with normal flow [54]. In our study, we noticed that NOCAD patients with angina-like chest pain did not differ statistically in terms of the value of this index with the control group.

Among other inflammatory parameters, CRP is one of the most common. It might affect coronary atherosclerosis development and reduce endothelial nitric oxide production [55,56,57]. Tangentially, increased CRP levels are associated with more frequent coronary angiographies and their complications in apparently healthy people [58,59,60]. Our study showed no significant alterations of the hs-CRP levels in the NOCAD group. Similar results were obtained in the American study, in which most NOCAD patients had no increase in CRP [61]. Nevertheless, several studies showed that 30–55% of NOCAD patients demonstrated a substantial increase in CRP levels [62,63,64], although CRP levels did not correlate with increased mortality rates [64,65]. Interestingly, higher CRP levels were also reported in patients with arterial hypertension [66] and endothelial dysfunction [61]. On the other hand, the study by Hung et al. revealed that women were more likely to suffer from coronary spasm at lower hs-CRP levels, while higher CRP levels were linked with coronary vasoconstriction in men [67]. Contrarily, another study revealed that NOCAD women had more increased CRP levels than men [68].

Cardiac troponins are the most specific and sensitive markers of myocardial injury. Their raised concentration reflects even small damage to the heart muscle but does not define its cause [69,70]. In our study, cTnI measurements revealed no significant increase in the NOCAD patients. Nevertheless, higher cTnI levels were observed in the OCAD group, although still below the cut-off point for MI. According to other studies, NOCAD patients may disclose a cTnI-levels increase [71] or its absence [72,73,74], the same as in our analysis. Moreover, higher troponin levels were observed in OCAD than in the NOCAD group [75,76,77], and correspondingly increased values of cTnI were more common in patients with OCAD (66%) than NOCAD (40%) [71].

Raised troponin levels in symptomatic patients are a substantial predictor of MACE [65]. Interestingly, patients with NOCAD who presented a significant increase in troponin due to a specific non-myocardial cause were at the same risk of death and MACE as patients with OCAD and elevated troponins during the 1-year follow-up [76]. On the other hand, NOCAD patients with an unidentified origin of raised troponins were in the low-risk group [75]. A study by Aldous et al. also showed that patients without CAD or with NOCAD, presenting with angina-like pain and elevated troponin values, were at low risk of death and heart attack during the 2-year follow-up [62]. Of note, this study excluded patients who had an increase in troponins due to non-cardiac reasons. Furthermore, about 80% of symptomatic NOCAD patients exhibited an abnormal coronary endothelial function and a significantly higher logarithm of a high-sensitive cTnI concentration than patients with normal endothelial function [78]. The cTnI levels greater than 12.5 ng/L showed a 100% specificity for predicting endothelial dysfunction [78].

Male gender is a well-known risk factor for CAD. However, studies show that angina-like chest pain in patients without hemodynamically significant stenoses of large coronary arteries mostly affects women [71,77,79]. This might be associated with coronary microvascular dysfunction, including endothelial impairment or spasm of epicardial arteries [71,79]. Our findings support a higher prevalence of NOCAD in women.

Furthermore, our study revealed that arterial hypertension is an independent risk factor of OCAD, increasing this risk almost five-fold. Conversely, NOCAD patients are much less likely to suffer from arterial hypertension. Our results are consistent with the studies in which most NOCAD patients did not have arterial hypertension [62,72]. However, in neither of these studies was the predominance of patients without arterial hypertension in the NOCAD group as significant as in our study. In contrast to our research, a few studies reported that most NOCAD patients presented with coexisting arterial hypertension [66,75] or that groups of patients with and without hypertension were comparable [71].

Several studies showed that DM, regardless of type, is a potent risk factor for CAD [80,81]. Additionally, patients suffering from DM are characterized by an increased MI incidence, higher post-MI mortality rates, and a more frequent recurrence of MI compared to patients without DM [82]. Because angina may be an indicator of coronary artery lesions [83], its absence or reduced sensation resulting from diabetic neuropathy [84], especially in elderly patients [85], may lead to a delayed diagnosis of CAD, possibly contributing to the increased post-MI mortality [86]. Furthermore, angiographic examinations revealed that silent ischemia is associated with much more advanced coronary lesions at the time of diagnosis than in patients without DM [87]. Our results indicate that DM increases the incidence of OCAD and is almost two times less common in NOCAD patients. Our work reinforces the evidence that there is a significant predominance of non-diabetic patients in the NOCAD group [62,66,71], although the role of DM in the pathogenesis of NOCAD should not be underestimated. A recent study revealed that the presence of DM is associated with CMD, and diabetic patients are characterized by significantly lowered Coronary Flow Reserve (CFR) and Microvascular Resistance Reserve (MRR) [88]. Furthermore, poor glycemic control was reported as an independent risk factor for CMD in the NOCAD population [89]. Interestingly, the results of our study did not show statistical significance for DM in multivariable regression models. This might be explained by the characteristics of the studied population, in which we have achieved a surprisingly high percentage of diabetic patients in general (33%) and especially in the NOCAD group. A total of 24% of the NOCAD individuals were diagnosed with DM, compared to 11, 12, or 18% observed in other studies [71,90,91]. This leads to an extraordinary low relation of diabetic patients in the OCAD to the NOCAD group in our study (1.5 vs. 2.1, 1.9, or 1.8 in the other studies) [71,90,91], which may influence the results of the statistical analyses. The cause of a relatively high percentage of diabetic patients in the NOCAD group may be at least partially connected with the older median age reported in our study, in comparison to previous research (66.0 vs. 62.6 or 57.0) [71,90,91].

Our study has several limitations. Data were analyzed retrospectively and some data regarding smoking habits and alcohol consumption are missing. Furthermore, it was difficult to confirm the pre-hospital diagnosis of dyslipidemia, and patient lipid levels prior to statin administration were also not available. Unluckily, over half of the patients lacked information about noninvasive diagnostic procedures performed before coronary angiography, such as treadmill exercise stress.

## 5. Conclusions

To conclude, female sex, a more abnormal lipid profile, and the lack of arterial hypertension and/or DM are the factors most highly associated with non-obstructive coronary arteries in angina patients undergoing coronary angiography. Higher TC, LDL-C, and non-HDL-C levels can be explained by less-intensive statins therapy, but further studies in this area are needed.

## Figures and Tables

**Figure 1 jpm-12-01049-f001:**
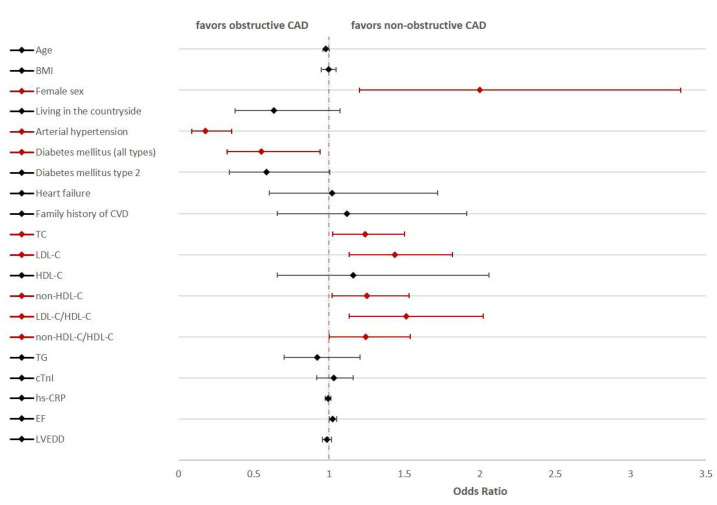
Univariable logistic regression analysis.

**Table 1 jpm-12-01049-t001:** Study population characteristics.

Parameter	NOCAD Group *N (% Value) or* *Median (q1–q3)*	OCAD Group *N (% Value) or* *Median (q1–q3)*	*p*-Value
Number of participants	136 (100%)	128 (100%)	
*Personal characteristics*			
Gender (females) **	61 (45%)	37 (29%)	*p* = 0.007
Living in the countryside/in the city	100/36 (74%/26%)	82/46 (64%/36%)	*p* = 0.088
Age	66.00 (59.00–73.00)	68.00 (61.00–74.50)	*p* = 0.149
BMI (kg/m^2^)	28.26 (24.84–31.48)	28.05 (25.49–31.89)	*p* = 0.607
*Comorbidities*			
Arterial hypertension ***	86 (63%)	116 (91%)	*p* < 0.001
Diabetes mellitus (all types) *	32 (24%)	46 (36%)	*p* = 0.027
DM2	31 (23%)	43 (34%)	*p* = 0.051
Heart failure	42 (31%)	39 (30%)	*p* = 0.941
Past myocardial infarction ***	13 (10%)	54 (42%)	*p* < 0.001
Past PCI (at least one) ***	12 (9%)	79 (62%)	*p* < 0.001
Family history of CVD	66 (60%) Total N: 110 (100%) *^a^*	63 (57%) Total N: 110 (100%) *^a^*	*p* = 0.681
*Addictions*			
*Alcohol consumption*	Total N: 73 (100%) *^a^*	Total N: 83 (100%) *^a^*	
No	57 (78%)	69 (83%)	*p* = 0.645
Occasionally	14 (19%)	13 (16%)	
Frequently	2 (3%)	1(1%)	
*Smoking*	Total N: 113 (100%) *^a^*	Total N: 109 (100%) *^a^*	
Never	60 (53%)	53 (49%)	*p* = 0.728
Active smoker	23 (20%)	22 (20%)	
Smoking in the past	30 (27%)	34 (31%)	

* *p* < 0.05; ** *p* < 0.01; *** *p* < 0.001; *^a^* in the case of family history of CVD, alcohol consumption, and smoking, the total N differs from the number of participants in each group due to the missing information in medical records. BMI—body mass index; CVD—cardiovascular disease; DM2—type 2 diabetes mellitus; NOCAD—non-obstructive coronary artery disease; OCAD—obstructive coronary artery disease; PCI—percutaneous coronary intervention.

**Table 2 jpm-12-01049-t002:** Laboratory and echocardiographic parameters.

Parameter	Unit	NOCAD Group *Median (q1–q3)*	OCAD Group *Median (q1–q3)*	*p*-Value
*Lipid parameters*				
TC *	mmol/L	4.40 (3.78–5.63)	4.12 (3.42–5.01)	*p* = 0.026
LDL-C **	mmol/L	2.32 (1.80–3.50)	2.10 (1.50–2.70)	*p* = 0.003
HDL-C	mmol/L	1.42 (1.15–1.76)	1.43 (1.14–1.68)	*p* = 0.580
Non-HDL-C *	mmol/L	2.89 (2.29–4.19)	2.66 (2.06–3.39)	*p* = 0.045
TG	mmol/L	1.17 (0.81–1.55)	1.18 (0.84–1.65)	*p* = 0.849
LDL-C/HDL-C ratio *	1	1.75 (1.22–2.60)	1.50 (1.10–1.95)	*p* = 0.018
Non-HDL-C/HDL-C ratio	1	2.18 (1.41–3.17)	1.93 (1.45–2.63)	*p* = 0.131
*White blood parameters and C-reactive protein*				
White blood count	10^9^/L	7.33 (6.12–8.89)	7.02 (5.87–8.95)	*p* = 0.452
Neutrophil count	10^9^/L	4.87 (3.76–6.04)	4.52 (3.66–6.08)	*p* = 0.481
Eosinophil count	10^9^/L	0.12 (0.06–0.18)	0.13 (0.07–0.21)	*p* = 0.214
Basophil count	10^9^/L	0.03 (0.02–0.05)	0.03 (0.02–0.04)	*p* = 0.277
Lymphocyte count	10^9^/L	1.55 (1.19–1.98)	1.60 (1.26–1.95)	*p* = 0.354
Monocyte count	10^9^/L	0.44 (0.35–0.54)	0.43 (0.34–0.53)	*p* = 0.492
NLR	1	3.14 (2.24–4.36)	2.84 (2.18–4.01)	*p* = 0.294
LMR	1	3.57 (2.69–4.72)	3.76 (2.80–4.60)	*p* = 0.416
hs-CRP	mg/L	2.90 (1.60–5.40)	5.15 (3.00–7.10)	*p* = 0.096
*Troponins*				
cTnI **	ng/mL	0.000 (0.000–0.007)	0.002 (0.000–0.015)	*p* = 0.004
*Echocardiographic parameters*				
EF ***	%	60.00 (55.00–60.00)	55.00 (50.00–60.00)	*p* < 0.001
LA diameter	mm	42.00 (38.00–46.00)	42.50 (39.00–47.00)	*p* = 0.464
LVEDD	mm	49.00 (45.00–54.00)	52.00 (47.00–56.00)	*p* = 0.059
RVD	mm	31.00 (29.00–36.00)	32.00 (30.00–35.00)	*p* = 0.384
IVS thickness	mm	11.00 (10.00–12.00)	11.00 (10.00–12.00)	*p* = 0.898
PW thickness	mm	11.00 (10.00–12.00)	11.00 (10.00–12.00)	*p* = 0.611

* *p* < 0.05; ** *p* < 0.01; *** *p* < 0.001; cTnI—cardiac troponin I; EF—ejection fraction; HDL-C—high-density lipoprotein cholesterol; hs-CRP—high-sensitive C-reactive protein; IVS—interventricular septum; LA—left atrium; LDL-C—low-density lipoprotein cholesterol; LMR—lymphocyte/monocyte ratio; LVEDD—left ventricle end-diastolic diameter; NLR—neutrophil/lymphocyte ratio; NOCAD—non-obstructive coronary artery disease; Non-HDL-C—non-high-density lipoprotein cholesterol; OCAD—obstructive coronary artery disease; PW—posterior wall; RVD—right ventricle diameter; TC—total cholesterol; TG—triglycerides.

**Table 3 jpm-12-01049-t003:** Multivariable logistic regression models.

Parameter	OR	95% CI	*p*-Value
*Model 1*			
Arterial hypertension ***	0.21	0.10–0.43	*p* < 0.001
Diabetes mellitus (all types)	0.64	0.35–1.16	*p* = 0.139
Female sex **	2.37	1.33–4.20	*p* = 0.003
LDL-C/HDL-C **	1.65	1.19–2.29	*p* = 0.003
*Model 2*			
Arterial hypertension ***	0.20	0.10–0.41	*p* < 0.001
Diabetes mellitus (all types)	0.62	0.34–1.13	*p* = 0.119
Female sex *	1.96	1.12–3.43	*p* = 0.018
LDL-C *	1.39	1.08–1.79	*p* = 0.010
*Model 3*			
Arterial hypertension ***	0.20	0.10–0.41	*p* < 0.001
Diabetes mellitus (all types)	0.62	0.34–1.12	*p* = 0.110
Female sex **	2.31	1.30–4.09	*p* = 0.004
Non-HDL-C/HDL-C *	1.36	1.06–1.75	*p* = 0.015
*Model 4*			
Arterial hypertension ***	0.20	0.10–0.40	*p* < 0.001
Diabetes mellitus (all types)	0.64	0.35–1.14	*p* = 0.131
Female sex *	1.96	1.13–3.40	*p* = 0.017
Non-HDL-C *	1.26	1.01–1.56	*p* = 0.039
*Model 5*			
Arterial hypertension ***	0.13	0.05–0.35	*p* < 0.001
Diabetes mellitus (all types)	0.67	0.31–1.42	*p* = 0.293
Female sex *	2.45	1.21–4.94	*p* = 0.013
Approximate atorvastatin dose *^a^* **	0.97	0.96–0.99	*p* = 0.003

* *p* < 0.05; ** *p* < 0.01; *** *p* < 0.001; *^a^* calculation based on [30]; CI—confidence interval; HDL-C—high-density lipoprotein cholesterol; LDL-C—low-density lipoprotein cholesterol; non-HDL-C—non-high-density lipoprotein cholesterol; OR—odds ratio.

**Table 4 jpm-12-01049-t004:** Summary of pre-hospital medications in NOCAD and OCAD groups (excluding lipid-lowering drugs).

Medicine	Number of Participants *N (% Value)* Total N: 253 *^a^*		*p*-Value
	NOCAD N: 128 (100%)	OCAD N: 125 (100%)	
ACE-I/ARB ***	78 (60.94%)	105 (84.00%)	*p* < 0.001
Acetylsalicylic Acid ***	57 (44.53%)	100 (80.00%)	*p* < 0.001
b-Blockers	97 (75.78%)	93 (74.40%)	*p* = 0.721
Ca-blockers *	32 (25.00%)	47 (37.60%)	*p* = 0.035
Diuretics (thiazides + loop diuretics)	67 (52.34%)	64 (51.20%)	*p* = 0.805
Metformin	21 (16.40%)	33 (26.40%)	*p* = 0.057
MRA	37 (28.91%)	28 (22.40%)	*p* = 0.223
Nitrates *	3 (2.34%)	11 (8.80%)	*p* = 0.026
Trimetazidine	14 (10.94%)	20 (16.00%)	*p* = 0.249
NOAC *	26 (20.31%)	14 (11.20%)	*p* = 0.044
VKA **	35 (27.34%)	16 (12.80%)	*p* = 0.004

* *p* < 0.05; ** *p* < 0.01; *** *p* < 0.001; *^a^* the total N differs from the number of participants in each group due to the missing information in medical records. ACE-I—angiotensin-converting enzyme inhibitors; ARB—angiotensin II receptor blockers; MRA—mineralocorticoid receptor antagonists; NOAC—novel oral anticoagulants; NOCAD—non-obstructive coronary artery disease; OCAD—obstructive coronary artery disease; VKA—vitamin K antagonists.

**Table 5 jpm-12-01049-t005:** Pre-hospital lipid-lowering therapy.

Lipid-Lowering Therapy	Number of Participants *N (% Value)* Total N: 253 *^a^* or *Median (q1–q3)*		*p*-Value
	NOCAD N: 128 (100%)	OCAD N: 125 (100%)	
No treatment ***	58 (45.31%)	21 (16.80%)	*p* < 0.001
Statin treatment ***	70 (54.69%)	104 (83.20%)	*p* < 0.001
+10 mg/d ezetimibe	1 (0.78%)	3 (2.40%)	*p* = 0.305
+fenofibrate *	1 (0.78%)	7 (5.60%)	*p* = 0.031
median dose (mg/d) *	267 (267–267)	215 (160–215)	*p* = 0.045
Atorvastatin	43 (33.59%)	53 (42.40%)	*p* = 0.149
median dose (mg/d) **	20 (20–40)	40 (20–40)	*p* = 0.001
Rosuvastatin ***	25 (19.53%)	48 (38.40%)	*p* < 0.001
median dose (mg/d)	20 (10–20)	20 (20–40)	*p* = 0.120
Simvastatin	2 (1.56%)	3 (2.40%)	*p* = 0.681
median dose (mg/d) ***	15 (10–20)	40 (40–40)	*p* = 0.007
All statins converted into approximate atorvastatin dose *^b^* ***	20 (20–40)	40 (20–40)	*p* < 0.001
>20 mg/d ***	30 (23.44%)	76 (60.80%)	*p* < 0.001
≤20 mg/d ***	98 (76.56%)	49 (39.20%)	*p* < 0.001

* *p* < 0.05; ** *p* < 0.01; *** *p* < 0.001; *^a^* the total N differs from the number of participants in each group due to the missing information in medical records. *^b^* calculation based on [30]. NOCAD—non-obstructive coronary artery disease; OCAD—obstructive coronary artery disease.

## Data Availability

The datasets used and/or analyzed during the current study are available from the corresponding author on reasonable request.

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
