# Peer review of "Risk Factors Predisposing to Angina in Patients with Non-Obstructive Coronary Arteries: A Retrospective Analysis"

_jpm, 2022, doi:10.3390/jpm12071049_

Round 1

Reviewer 1 Report

Wiśniewski et al. aimed to explore Risk Factors Predisposing to Angina in Patients with Non-Obstructive Coronary Arteries. The present study is based on valid idea, but it was poorly executed. For such a study design to be relevant, the sample would have to be much bigger in order to adjust for all the confounders. I wonder if the authors calculated the sample size in the first place? The other big issue is novelty of the present paper. All these predisposing factors were assessed in large populational studies.

On the other hand, creation of risk score model and its validation would perhaps make this paper valuable.

There are also other minor remarks that would require attention such as the fact that materials and methods section which was written very cursory and that there are leftovers from template in the results section.

Author Response

Dear Reviewer,

the authors would like to thank you for valuable comments, which helped to improve the quality of the manuscript.

Please find our detailed revision note below.

Sincerely yours,

Authors

Reviewer 1

Wiśniewski et al. aimed to explore Risk Factors Predisposing to Angina in Patients with Non-Obstructive Coronary Arteries. The present study is based on valid idea, but it was poorly executed. For such a study design to be relevant, the sample would have to be much bigger in order to adjust for all the confounders. I wonder if the authors calculated the sample size in the first place?

Answer:

Based on previously published research papers [1,2], we initially aimed to include 277 patients in each group. This sample size would be appropriate to achieve a statistical power of over 80% when assessing the differences in statin use between the groups if we make an assumption that our population would fit the same percentage of statin users and non-users in both the NOCAD and OCAD group. This calculation assumed the 5% significance level, using a Z-test to compare two different groups.

However, while analyzing the tentative results during data collection, the outcomes regarding lipid profile appeared so thought-provoking that we decided to publish them. With our paper, we would like to emphasize that prescribing lipid-lowering drugs in the NOCAD group is still underestimated, although it may influence the prognosis and the risk of MACE. Since our results may help improve awareness of the appropriate management of dyslipidemia in the NOCAD population, we feel it is also ethically essential to share our preliminary results with clinicians. Before submission, we have calculated statistical power for statistically significant lipid profile differences, the prevalence of statin use, as well as the mean dose of statin administered (converted into approximate equivalent atorvastatin dose), which are the key findings of our study. We have obtained a statistical power of over 80% (0.824–0.998, depending on a selected parameter), which supports that our results are credible. Our future perspectives include completion of the initially planned sample size as well as starting a prospective study with a follow-up.

Of course, we agree that a large populational study would be the best option to test our hypotheses. However, for technical and economic reasons performing such a large study is currently unavailable in our institution. Nevertheless, we believe that our paper may highlight the problem of poor management of hypercholesterolemia in the NOCAD group and inspire further research.

References:

[1] Pizzi C, Xhyheri B, Costa GM, et al. Nonobstructive Versus Obstructive Coronary Artery Disease in Acute Coronary Syndrome: A Meta-Analysis. J Am Heart Assoc. 2016;5(12):e004185. Published 2016 Dec 16. doi:10.1161/JAHA.116.004185.

[2] Dasari TW, Golwala H, Koehler M, Wayangankar S, Pakala A, Schechter E, Lozano P, Abu-Fadel MS, Latif F, Thadani U. Is risk factor control and guideline-based medical therapy optimal in patients with nonobstructive coronary artery disease? A Veterans Affairs study. Am J Med Sci. 2013 May;345(5):339-42. doi: 10.1097/MAJ.0b013e31825c6951.

The other big issue is novelty of the present paper. All these predisposing factors were assessed in large populational studies. On the other hand, creation of risk score model and its validation would perhaps make this paper valuable.

Answer: The novelty of our paper is the evidence that higher lipid parameters could be paradoxically associated with NOCAD instead of OCAD, which may result from poor control of hypercholesterolemia in the NOCAD group. These results were obtained in the Polish population, but we believe that there is still a lot to emphasize in the context of management of dyslipidemia and cardiovascular risk factors in general worldwide and that the significance of lipid-lowering drugs in NOCAD patients is frequently underestimated. Although the number of patients in both groups is low, when compared to populational studies, we have reached appropriately high statistical power (0.998) to recognize the results as credible.

In line with the suggestion of Reviewer 2, we have also performed an additional analysis within the OCAD group regarding patients with obstruction in one or multiple coronary arteries. The results of the analysis are presented in the Supplementary material in Table 2 and may also improve the value of our paper.

Unluckily, we were not able to create and validate a risk score model in our institution, and present it in the ongoing manuscript. Nevertheless, we would definitely use your suggestion in our further publications. Based on the results of the current work, we are going to prepare a prospective study focused on creating a score model.

There are also other minor remarks that would require attention such as the fact that materials and methods section which was written very cursory and that there are leftovers from template in the results section.

Answer: Since a large part of the information regarding the study population was presented in the form of a table (Table 1), we decided not to replicate this information in the material and methods section. Now, we have clearly indicated in the material and methods section that some information regarding the study population is summarized in Table 1. Furthermore, in line with your suggestion, the material and methods section has been rewritten and supplemented with missing information to provide the most comprehensive outlook. In addition, the whole manuscript has been carefully checked, and writing mistakes, including template leftovers, have been corrected.

Reviewer 2 Report

The current manuscript examines the risk factors associated with non-obstructive coronary artery disease (NObCAD) in a retrospective analysis. Using multivariable regression model, the authors tried to demonstrate that female sex as the independent risk factor of NObCAD, and that arterial hypertension substantially lowered the probability of NObCAD. The current investigation showed that increased TC, LDL-C, non-HDL, and LDL-C/HDL-C ratio are risk factors of NObCAD in patients reporting angina, along with female sex and the absence of arterial hypertension. This may result from poor control of hypercholesterolemia. Overall, the retrospective analysis was properly performed and their observation is clinically sound. However, some concerns listed below limit the clear narrative of the current study. 

1. In this study, the authors used univariable regression model as well as multivariable regression model to identify the key factors associated with NObCAD in patients. Did the authors try to perform further analysis for the control group (ObCAD) between the patients with lesions for one coronary artery and the patients with lesions for multiple coronary arteries? I wonder if there are difference results for these analysis between these subgroups for ObCAD. 

2. DM has been recognized as an independent risk factor for coronary artery disease. However, in the current study, the authors showed that DM was not an independent factor for NObCAD occurrence, which is an interesting result. The authors should further elaborate and discuss this results in the Discussion part. 

3. The English language used in this manuscript should be improved before consideration of publication.

4. Proof reading is needed:

1)    B-Blockers (should be b, not B) in Table 4. 

2)    Unproper paragraphing: Line 355-356.

Author Response

Dear Reviewer,

the authors would like to thank you for valuable comments, which helped to improve the quality of the manuscript.

Please find our detailed revision note below.

Sincerely yours,

Authors

Reviewer 2

The current manuscript examines the risk factors associated with non-obstructive coronary artery disease (NObCAD) in a retrospective analysis. Using multivariable regression model, the authors tried to demonstrate that female sex as the independent risk factor of NObCAD, and that arterial hypertension substantially lowered the probability of NObCAD. The current investigation showed that increased TC, LDL-C, non-HDL, and LDL-C/HDL-C ratio are risk factors of NObCAD in patients reporting angina, along with female sex and the absence of arterial hypertension. This may result from poor control of hypercholesterolemia. Overall, the retrospective analysis was properly performed and their observation is clinically sound. However, some concerns listed below limit the clear narrative of the current study.

  1. In this study, the authors used univariable regression model as well as multivariable regression model to identify the key factors associated with NObCAD in patients. Did the authors try to perform further analysis for the control group (ObCAD) between the patients with lesions for one coronary artery and the patients with lesions for multiple coronary arteries? I wonder if there are difference results for these analysis between these subgroups for ObCAD.

Answer: In line with your suggestion, an additional analysis regarding patients with lesions in one or multiple coronary arteries has been performed. The results of the analysis are presented in the Supplementary material in Tables S2-S5.

We found that median BMI was significantly higher in the group with only one obstructed coronary artery and multivariable regression analysis revealed that BMI could be recognized as an independent risk factor, decreasing the risk for multiple lesions (OR=0.908, 95%CI: 0.832 – 0.990; p=0.030). This unexpected result might fit into the pattern that the lower severity of CAD, the more underestimation from the patients and doctors it has, same as we presented in the case of lipid profile in our work. Furthermore, the past myocardial infarction or past PCI were unsurprisingly reported more frequently in the multiple obstructed coronary arteries group and were both considered the independent risk factors (OR=2.433 and OR=3.084, respectively). Notably, we observed that the use of mineralocorticoid receptor antagonists (MRA) was more common in the group with multiple coronary arteries obstructed, albeit the were no statistically significant differences in the prevalence of heart failure or hyper-tension. Interestingly, statin therapy did not differ between the groups.

  1. DM has been recognized as an independent risk factor for coronary artery disease. However, in the current study, the authors showed that DM was not an independent factor for NObCAD occurrence, which is an interesting result. The authors should further elaborate and discuss this results in the Discussion part.

Answer: In line with your suggestion, the fact that DM did not reach statistical significance to become an independent risk factor for NOCAD was extensively discussed in the manuscript (lines 407–417):

“Interestingly, the results of our study did not show statistical significance for DM in multivariable regression models. This might be explained by the characteristics of the studied population, in which we have achieved a surprisingly high percentage of diabetic patients in general (33%) and especially in the NOCAD group. 24% of the NOCAD individuals were diagnosed with DM, compared to 11%, 12%, or 18% observed in other studies [70,89,90]. This leads to extraordinary low relation of diabetic patients in the OCAD to the NOCAD group in our study (1.5 vs. 2.1, 1.9, or 1.8 in the other studies) [70,89,90], which may influence the results of statistical analyses. The cause of a relatively high percentage of diabetic patients in the NOCAD group may be at least partially connected with the older median age reported in our study, in comparison to previous research (66.0 vs. 62.6 or 57.0) [70,89,90].”.

  1. The English language used in this manuscript should be improved before consideration of publication.

Answer: The language of the manuscript has been corrected by an English native speaker before the submission. In order to follow your suggestion, the revised manuscript has been corrected once again by another native speaker. We believe that English is now appropriate for publication.

  1. Proof reading is needed:

1)    B-Blockers (should be b, not B) in Table 4.

2)    Unproper paragraphing: Line 355-356.

Answer: The suggested changes have been applied to the manuscript. In addition, the text has been carefully checked and another few writing mistakes have been corrected.

Reviewer 3 Report

The author should change the acronymus of NObCAD into NOCAD. 

In the introduction a citation about novel findings regarding the prevalence of CMD is missing  (35301851)

In the discussion is controversial the role of DM in the pathogenesis of NOCAD. Also in this case recent findings (34738020) suggest the a worse microvascular function in patients with DM compared to those with no DM. 

Author Response

Dear Reviewer,

the authors would like to thank you for valuable comments, which helped to improve the quality of the manuscript.

Please find our detailed revision note below.

Sincerely yours,

Authors

Reviewer 3

The author should change the acronymus of NObCAD into NOCAD.

Answer: The whole text has been carefully checked and corrected as suggested. The acronym NOCAD is now used throughout the manuscript. Consequently, the acronym “ObCAD” has been changed into “OCAD” throughout the manuscript.

In the introduction a citation about novel findings regarding the prevalence of CMD is missing (35301851)

Answer: The introduction has been rewritten and the suggested reference has been included.

In the discussion is controversial the role of DM in the pathogenesis of NOCAD. Also in this case recent findings (34738020) suggest the a worse microvascular function in patients with DM compared to those with no DM.

Answer: The part of the discussion regarding the role of diabetes mellitus in the pathogenesis of NOCAD has been corrected and the suggested reference has been included.

Round 2

Reviewer 1 Report

I thank the author for the comprehensive response, yet I still can find no benefit of the present study for the scientific community. The authors state that lipid lowering therapy is underappreciated in NOCAD, I agree, but that is not because this relationship is not explored, but perhaps more because of ignorance or whatever is the reason for not using well established data in clinical medicine. In fact, as I previously stated there are studies including more than 2000 participants with MINOCA describing independent correlation with MACEs in these patients. One may ask, do we really need additional evidence? Furthermore, the authors mention "paradoxical" association between NOCAD and lipoproteins. To be honest, I see no paradox, as relationship between endothelial dysfunction and lipoproteins is perhaps one of the most extensively studied pathophysiological relationship in medicine.

Author Response

Reviewer 1

I thank the author for the comprehensive response, yet I still can find no benefit of the present study for the scientific community. The authors state that lipid lowering therapy is underappreciated in NOCAD, I agree, but that is not because this relationship is not explored, but perhaps more because of ignorance or whatever is the reason for not using well established data in clinical medicine. In fact, as I previously stated there are studies including more than 2000 participants with MINOCA describing independent correlation with MACEs in these patients. One may ask, do we really need additional evidence? Furthermore, the authors mention "paradoxical" association between NOCAD and lipoproteins. To be honest, I see no paradox, as relationship between endothelial dysfunction and lipoproteins is perhaps one of the most extensively studied pathophysiological relationship in medicine.

Dear Reviewer,

thank you for your valuable comments. Of course, we agree that dyslipidemia is one of the crucial components causing endothelial dysfunction. However, this is not what we studied. Please, pay attention that in our paper we compared the NOCAD and the OCAD groups. We did not compare the NOCAD group to healthy individuals in order to see if lipoproteins are changed in the NOCAD. Our results showed that the lipid profile is more disturbed in the NOCAD group when compared to the OCAD. In our opinion, this is an unexpected finding. Looking into the pathogenesis of OCAD we suspected a worse lipid profile in the OCAD than in the NOCAD. Furthermore, in the discussion section, we extensively presented the literature data, which reported that lipid profile was more deteriorated in the OCAD group than in the NOCAD; we included inter alia the APPROACH and the CRUSADE trails as well as a large meta-analysis by Pizzi et al. performed at 120 000 patients. Taken together, we obtained just the opposite results regarding lipoproteins concentration than we should expect based on the pathophysiology and the current literature, and that is why we call them unexpected.

Since one may misconceive a sentence from the abstract regarding the study population, we have rewritten the fragment “[…] we analyzed medical records of consecutive 136 NOCAD patients and 128 controls with significant stenosis in at least one coronary artery.”, and replaced the word “controls” with “patients” to avoid ambiguity. We have also improved the rest of the abstract to give a clear message.

As for the literature evidence for the less frequent statin use in the NOCAD group, we agree that there are studies confirming this phenomenon. However, please, note that even though less frequent statin use in the NOCAD group was observed, those studies did not report concomitant changes in the lipid profile [34,35,1*,2*]. Although the less frequent statin use was shown, the differences in the lipid profile were not statistically significant between the NOCAD and OCAD groups. More precisely, in the vast majority of the research, lipid profile was still worse in the OCAD group despite the lower frequency of statin use, or the lipoproteins concentration were comparable between the NOCAD and OCAD groups with even no tendency observed. This may indicate that the discrepancy between the statin receiving and non-receiving patients was much bigger in our study populations than in previously published papers, so that it translated into a worse lipid profile in the NOCAD group, and should be acknowledged as a novelty of our study.

Our findings are of particular interest in personalizing the treatment strategy in the Polish population and presumably in other Slavic countries. To our best knowledge, there is no PubMed-indexed paper presenting both more disturbed lipid profile and less intensive statin therapy in the NOCAD group. It seems that the worldwide underestimation of statin therapy in the NOCAD group is particularly expressed in the Polish population. In our opinion, every effort should be made to emphasize the role of adequate statin therapy in the NOCAD. Publishing scientific data could serve as valuable background for convincing the patients and doctors to improve their statin treatment. This is why we also plan to start a prospective study with a follow-up to broaden the field and analyze the outcomes in OCAD, NOCAD, and NOCAD subpopulations.

Sincerely yours,

Authors

[33] Choo, E.H.; Chang, K.; Lee, K.Y.; Lee, D.; Kim, J.G.; Ahn, Y.; Kim, Y.J.; Chae, S.C.; Cho, M.C.; Kim, C.J.; et al. Prognosis and Predictors of Mortality in Patients Suffering Myocardial Infarction With Non-Obstructive Coronary Arteries. J Am Heart Assoc 2019, 8, e011990, doi:10.1161/JAHA.119.011990.

[34] Ballesteros-Ortega, D.; Martínez-González, O.; Gómez-Casero, R.B.; Quintana-Díaz, M.; de Miguel-Balsa, E.; Martín-Parra, C.; López-Matamala, B.; Chana-García, M.; Alonso-Fernández, M.Á.; Manso-Álvarez, M. Characteristics of Patients with Myo-cardial Infarction with Nonobstructive Coronary Arteries (MINOCA) from the ARIAM-SEMICYUC Registry: Development of a Score for Predicting MINOCA. Vasc Health Risk Manag 2019, 15, 57–67, doi:10.2147/VHRM.S185082.

[1*] Turgeon, R.D.; Sedlak, T. Use of Preventive Medications in Patients With Nonobstructive Coronary Artery Disease: Analysis of the PROMISE Trial. CJC Open. 2020; 3, 159-166. doi: 10.1016/j.cjco.2020.09.022.

[2*] Durmaz, E.; Ikitimur, B.; Karadag, B.; Barman, H.A.; Atici, A.; Koca, D.; Raimoglu, U.; Karaca, O.F.; Mutlu, D.; Ongen, Z. The impact of atherosclerotic risk factors on disease progression in patients with previously diagnosed nonobstructive coronary artery disease: factors affecting coronary artery disease progression. Coron Artery Dis. 2020; 31, 365-371. doi: 10.1097/MCA.0000000000000839.

Reviewer 2 Report

The authors have properly addressed my concerns in the revised manuscript. I have no further comments.

Author Response

Reviewer 2

The authors have properly addressed my concerns in the revised manuscript. I have no further comments.

Dear Reviewer,

thank you for your appreciation!

Sincerely yours,

Authors

Reviewer 3 Report

The manuscript has been appropriately revised. However the suggested reference (35301851) of the recently published meta analysis about CMD has not been added. 

I have no further comments. 

Author Response

Reviewer 3

The manuscript has been appropriately revised. However the suggested reference (35301851) of the recently published meta analysis about CMD has not been added.

Dear Reviewer,

thank you for your valuable comment. In the revised manuscript we have added the suggested reference [16] to improve the quality of the introduction.

Sincerely yours,

Authors

[16] Mileva, N.; Nagumo, S.; Mizukami, T.; Sonck, J.; Berry, C.; Gallinoro, E.; Monizzi, G.; Candreva, A.; Munhoz, D.; Vassilev, D.; et al. Prevalence of Coronary Microvascular Disease and Coronary Vasospasm in Patients With Nonobstructive Coronary Artery Disease: Systematic Review and Meta-Analysis. J Am Heart Assoc 2022, 11, e023207, doi:10.1161/JAHA.121.023207.
